# Emergency department attendance by callers to NHS111 who end the call prior to triage: A time-to-event-analysis

Richard Pilbery[1,2]*, Jen Lewis[2], Rebecca Simpson[2]

1 Yorkshire Ambulance Service Research Institute, Yorkshire Ambulance Service NHS Trust, Wakefield, United Kingdom, 2 Sheffield Centre for Health and Related Research, Division of Population Health, School of Medicine and Population Health, University of Sheffield, Sheffield, England

☯ These authors contributed equally to this work.
* r.s.pilbery@sheffield.ac.uk

## Abstract

### Background

The English National Health Service (NHS) 111 telephone service aims to assist members of the public with urgent medical care needs. However, each year nearly 18% of the 20.6 million calls to NHS 111 are abandoned prior to speaking to a health advisor. There are concerns that callers who are not triaged may not appropriately seek the correct level of care for their needs.

The aim of this study was to explore the patient journey for callers who contact NHS 111 but end the call prior to speaking to a health advisor. The primary objective was to determine whether ending an NHS 111 call prior to triage impacts the time taken for a patient with urgent healthcare needs to attend an Emergency Department (ED).

### Methods

We obtained routine data pertaining to all NHS 111 calls made by adult patients registered with a General Practitioner (GP) in the Bradford region of Yorkshire, England, between the 1st January 2022 and 30th June 2023. Subsequent healthcare access in the 72 hours following each caller's first (index) call was identified using the Connected Bradford research database.

We conducted a time-to-event analysis comparing the two cohorts: those whose index call was triaged by an NHS 111 health advisor vs. callers who ended the index call prior to triage. The 'event' was defined as an ED attendance within 72 hours for a non-avoidable cause.

We utilised Kaplan–Meier (KM) curves and conducted log-rank tests to compare the time to first non-avoidable ED attendance between cohorts, and a Cox proportional hazards model adjusted for baseline characteristics. From this, we calculated the adjusted hazard ratio (aHR) of attending an ED with a non-avoidable cause.

**Data availability statement:** Data cannot be shared publicly because the study dataset was derived from Connected Yorkshire research database data, which has strict controls on access as part of the ethical approvals by the NHS Health Research Authority, Research Ethics Committee and Confidentiality Advisory Group that are in place for the database. However, the data are available for researchers who meet the criteria for access to confidential data, by making an application to the Bradford Institute for Health Research (contact via email: bradfordresearch@bthft.nhs.uk).

**Funding:** This study is independent research funded by the National Institute for Health and Care Research, Yorkshire and Humber Applied Research Collaborations NIHR200166. The funders had no role in study design, data collection and analysis, decision to publish, or preparation of the manuscript.

**Competing interests:** The authors have declared that no competing interests exist.

## Results

There were 19,056 index non-triaged and 168,609 triaged calls made to NHS 111 by an adult registered with a Bradford GP. A higher proportion of ED attendances in the triaged call cohort were non-avoidable compared with the non-triaged cohort (84.6% compared to 80.0% for triaged calls). In addition, callers in the non-triaged call cohort attended ED later than the triaged call cohort (median 10 vs 8 hours, $p < 0.001$ by log rank test). The aHR for non-triaged calls vs triaged calls was 0.32 (95%CI 0.30–0.34).

## Conclusion

The time-to-event analysis found that callers to NHS 111 who do not wait to be triaged are slower to attend ED with a non-avoidable cause than those who are triaged, and are more likely to attend ED with an avoidable cause than triaged callers. This suggests that, for patients with a serious health problem that would be considered non-avoidable at ED, triaging by NHS 111 was associated with a reduced time to ED attendance.

## Introduction

The English National Health Service (NHS) 111 service is a telephone triage service, which aims to assist members of the public with urgent medical care needs and is the successor to the NHS Direct service in England. Its key founding objective was to provide easy access to support for the public with urgent care needs, to ensure they received the "right care, from the right person, in the right place, at the right time" [1]. It is also the key component of the "24/7 Integrated Urgent Care Service" outlined in the NHS Long Term Plan, acting as a "single point of access" for urgent care to facilitate co-ordinated access to services, including signposting to alternative care pathways and in doing so, assist in the provision of timely and appropriate care, an improved patient experience and reduce pressure on EDs [2].

However, in 2022, nearly 3.7 million callers to NHS 111 ended the call prior to speaking to a health advisor. This represents nearly 18% of the 20.6 million calls to NHS 111 each year [3] and has raised concerns about callers with urgent care needs not receiving timely care and advice [4]. While the scale of the issue has been quantified, there appears to be no research exploring to the healthcare trajectory or health outcomes of callers who are unable to speak to an NHS 111 health advisor, or why callers do not wait to be triaged, although the delay in answering calls has been mooted as a factor [4]. In addition, there are concerns that callers may seek alternative healthcare services, such as the ambulance service and emergency departments (EDs), but there is no evidence to confirm or refute this.

The aim of this study was to compare the patient journey for callers who contact NHS 111 but end the call prior to speaking to a health advisor, and those who are triaged. The primary objective was to determine whether ending an NHS 111 call prior

to triage impacts the time taken for a patient with urgent healthcare needs to attend ED. The secondary objective was to determine whether ending an NHS 111 call prior to triage impacts the time taken for a patient to attend ED regardless of urgency.

## Methods

### Data

We obtained routine, retrospective data from the Connected Bradford research database, which provides pseudonymised linked data for approximately 1.2 million citizens across the Bradford and Airedale region of Yorkshire [5]. Datasets include NHS 111 and 999 call data (including abandoned calls to NHS 111 since 2022), as well as primary and secondary care (including ED and in-patient activity for Bradford Royal Infirmary, Calderdale Royal Infirmary and Airedale General Hospital). All datasets are pseudonymised so that researchers cannot identify individual participants.

A supplemental dataset containing details of callers who had contacted NHS 111, ended the call after 30 seconds but prior to being triaged by an NHS 111 health advisor, was collated and provided by analysts from Yorkshire Ambulance Service NHS Trust (YAS) to the Connected Bradford research database. They identified these callers by examining triaged NHS 111 call records in the 12 months prior to, and 1 month post, index call. Approximately 80% of these non-triaged calls were able to be matched to an NHS number in this way.

The dataset for analysis comprised all NHS 111 calls made by adult (18 years and over) patients registered with a General Practitioner (GP) in the Bradford area at the time of the call between the 1st January 2022 and 30th June 2023. Index NHS 111 calls were identified as those made by patients who had not had been triaged by NHS 111 in the 72 hours prior to the first (index) call. Subsequent healthcare system access in the following 72 hours after the index call (whether triaged or not) was identified by searching the NHS 111 and 999 call, primary care, and hospital emergency department and in-patient admission datasets.

### Analysis

We conducted descriptive and time-to-event analyses comparing the two cohorts (those triaged by an NHS 111 call handler vs callers who ended the call prior to triage). The 'event' was defined as an ED attendance within 72 hours for a non-avoidable cause as defined by O'Keeffe et al. [6]. They defined an avoidable attendance as a patient presenting to a consultant-led ED which provides a 24-hour service with full resuscitation facilities and designated accommodation for the reception of emergency care patients (referred to as a type 1 ED [7]), but who do not receive investigations, treatments or referral that required the facilities of that ED. In consultation with the original lead author of the avoidable attendance paper and an experienced ED consultant/clinical academic, several additional discharge codes were categorised as indicative of a potentially avoidable admission (S1 Table).

### Primary analysis

For the primary outcome analysis we utilised Kaplan–Meier (KM) curves and conducted a log-rank test, to compare the time to first non-avoidable ED attendance and determine whether there was a significant difference in unadjusted time-to-event between cohorts. In addition, a Cox proportional hazards model was used to adjust for clustering of results by caller, and for baseline characteristics that have been implicated as potentially affecting the outcome, including age, sex, ethnicity, whether the call was in- or out- of-hours, and index of multiple deprivation (IMD) [8,9]. This enabled us to calculate the adjusted hazard ratio (aHR) of attending an ED with a non-avoidable cause for callers who ended the call prior to triage, compared to those who were triaged by an NHS 111 call handler. All analysis was conducted using the statistics package, R (v4.2.1), and utilised the ggsurvfit, survival and coxph packages to generate KM plots, log-rank and the Cox regression, respectively.

### Secondary analysis

The secondary outcome analysis was conducted as for the primary outcome, except the event outcome was ED attendance for any cause.

### Proportional hazard assumptions

A key consideration when undertaking a time-to-event analysis using Cox regression is ensuring that models conform to the proportion hazards assumption [10]. We checked for violations of the proportional hazards assumption by plotting log-log plots, Schoenfeld and Martingale residual plots and testing goodness-of-fit using the method described by Grambsch & Therneau [11] Our initial models violated these assumptions. To mitigate this, we adjusted the model by using time-splitting into hourly segments, and stratifying the analysis by the time of call (in-hours vs out-of-hours). While this improved model fit, both the cohort variable (non-triaged or triage call) and patient sex continued to violate the proportional hazards assumption to a small degree.

### Ethical approval

This study was approved by the Bradford Learning Health System Board in accordance with the Connected Yorkshire NHS Research Ethics Committee (REC) approval relating to the Connected Yorkshire research database (17/EM/0254). As a research database, Connected Yorkshire have REC and Confidential Advisory Group (CAG) approval for use of the data without individual patient consent. No separate Health Research Authority (HRA) approval was required for this study.

This study is independent research funded by the National Institute for Health and Care Research, Yorkshire and Humber Applied Research Collaborations NIHR200166. The funders had no role in study design, data collection and analysis, decision to publish, or preparation of the manuscript.

### Patient and public involvement

The application and protocol for this study was reviewed by the YAS patient research ambassador. In addition, Connected Bradford have an active patient and public involvement group who were involved in the decision to approve this study.

## Results

Between the 1st January 2022 and 30th June 2023 there were 19,056 index non-triaged calls and 168,609 triaged calls to NHS 111; non-triaged calls comprised approximately 10% of all index calls made by an adult registered with a Bradford GP (Table 1). A higher proportion of ED attendances in the triaged call cohort were non-avoidable compared with the non-triaged cohort (84.6% compared to 80.0% for triaged calls). In addition, callers in the non-triaged NHS 111 cohorts attended ED later than the triaged call cohort (median 10 vs 8 hours and 9 vs 7 hours for non-avoidable and all ED attendances, respectively).

### Kaplan-Meier plots

Both the Kaplan-Meier plots (Fig 1) and log-rank tests suggest that there is a significant difference between the non-triaged and triaged call cohorts for non-avoidable, and all, ED attendances (log-rank test p<0.001 for both non-avoidable and all ED attendances). This unadjusted result suggests that non-triaged callers are less likely than triaged callers to attend ED for non-avoidable, and any, cause.

### Cox regression

The aHR from the Cox regression suggests that non-triaged callers who have not yet attended ED for a non-avoidable cause, are around a third as likely to do so in the next hour compared to a triaged caller, for the 72-hour period following an index 111 call (Table 2, S1 File).

**Table 1. Summary of study data stratified by grouping.**

| Characteristic | Non-triaged Call (n = 19,056) | Triaged 111 Call (n = 168,609) |
|---|---|---|
| **Out-of-hours, n (%)** | | |
| In-hours | 5,092 (27%) | 58,687 (35%) |
| Out-of-hours | 13,964 (73%) | 109,922 (65%) |
| **Age in years, Median (IQR)** | 37 (28, 57) | 41 (28, 61) |
| **Sex, n (%)** | | |
| Female | 12,289 (64%) | 100,814 (60%) |
| Male | 6,766 (36%) | 67,791 (40%) |
| Unknown | 1 | 4 |
| **Index of Multiple Deprivation quintile, n (%)** | | |
| 1 (Most deprived) | 10,128 (55%) | 80,423 (50%) |
| 2 | 3,721 (20%) | 33,558 (21%) |
| 3 | 1,941 (11%) | 18,990 (12%) |
| 4 | 1,531 (8.4%) | 16,506 (10%) |
| 5 (Least deprived) | 993 (5.4%) | 12,307 (7.6%) |
| Unknown | 742 | 6,825 |
| **Ethnicity, n (%)** | | |
| White | 8,485 (45%) | 81,319 (48%) |
| Asian or Asian British | 5,317 (28%) | 37,829 (22%) |
| Black or African or Caribbean or Black British | 260 (1.4%) | 2,928 (1.7%) |
| Mixed multiple ethnic groups | 275 (1.4%) | 2,228 (1.3%) |
| Other ethnic group | 295 (1.5%) | 2,559 (1.5%) |
| Unknown/Refuse to say | 4,424 (23%) | 41,746 (25%) |
| **Index calls with one or more Emergency Department attendance, n** | 1,375 (7.2%) | 35,505 (21%) |
| **Index calls with one or more non-avoidable Emergency Department attendance, n (%)** | 1,099 (5.8%) | 30,070 (18%) |
| **Median time to Emergency Department attendance in hours, (IQR)** | 9 (5, 22) | 7 (4, 14) |
| **Median time to non-avoidable Emergency Department attendance in hours, (IQR)** | 10 (5, 23) | 8 (5, 15) |

## Discussion

As far as we are aware, this is the first study to examine the healthcare trajectory for callers to NHS 111 who end the call prior to triage. The time-to-event analysis suggests that callers who end their NHS 111 call prior to triage and who have not yet attended ED for a non-avoidable cause within 72 hours following the index call, are around a third as likely to do so in the next hour compared to a triaged caller. In addition, non-triage by NHS 111 was associated with later attendance at ED when compared with triaged callers.

There are also several notable differences between the groups. The proportion of calls that occured out-of-hours is higher in the non-triaged call group and both the IMD and ethnicity show differences, with a higher proportion of callers in the non-triaged cohort living in the most deprived quintile. It is not clear from the data why that should be, but may indicate a health inequality issue in line with previous evidence surrounding the difficulties in accessing services by people living in poverty, for example [12]. While there are differences in ethnicity, poor recording of this variable (it is missing in almost a quarter of cases), means it is difficult to comment further [13].

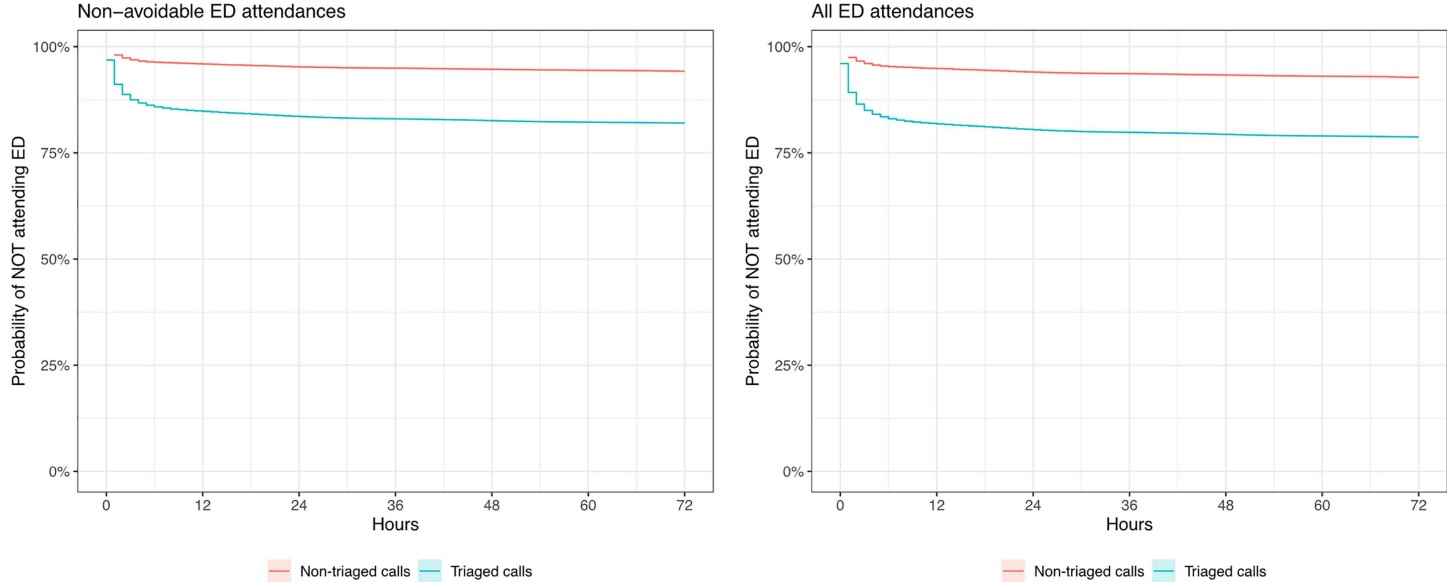

**Fig 1. Kaplan-Meier plots stratified by primary and secondary outcome.** Note: confidence intervals are present in the figure, but are narrow.

**Table 2. Results of Cox regression.**

| Abandoned calls (N) | Triaged 111 calls (N) | Non-avoidable ED attendances | | All ED attendances | |
|---|---|---|---|---|---|
| | | Unadjusted HR (95% CI) | Adjusted HR (95% CI) | Unadjusted HR (95%CI) | Adjusted HR (95%CI) |
| 19,056 | 168,609 | 0.3 (0.28–0.32) | 0.32 (0.3–0.34) | 0.31 (0.3–0.33) | 0.33 (0.31–0.35) |

Note: reference level for hazard ratios are triaged 111 calls.

Since by definition the non-triaged callers were not triaged, it is not possible to determine the acuity level that would have been assigned to them by NHS 111.

For callers who were triaged however, the performance of NHS 111 seems reasonable, with nearly 85% of triaged cases attending ED for a non-avoidable cause. However, we do not know whether this was directly as a result of being advised to attend by NHS 111, or the patient deciding for themselves. A previous study by Lewis et al [8] for example, demonstrated that patients do not always follow the advice provided by NHS 111 and attend ED even if it that was not the triage disposition reached. In addition, they found that in a number of cases where attendance at ED had not been indicated by NHS 111, the patient subsequently required admission.

Based on the results of this study, we would recommend that efforts are made to reduce the proportion of calls that end prior to triage, particularly with respect to the time taken to answer calls, since this is within the remit of the NHS 111 service to address. There may be other reasons that we could not measure in this study relating to why callers ended prior to triage, but it is reasonable to assume that the longer a caller is made to wait for triage, the more likely they will end the call before it is answered.

## Strengths and limitations

This study has described the demographic and healthcare system access characteristics of a population who are challenging to identify. We have also highlighted that this group may delay attending ED despite having a presentation that

warrants attendance. However, we are unable to determine whether there are longer term consequences of delayed (or non-) attendance at ED relating to the reason they contacted NHS 111. We also have no way of knowing whether these callers utilised NHS 111 Online instead, since granular patient level data is not available in the Connected Bradford data-set for NHS 111 Online access.

While the non-triaged call data did successfully identify the caller in most cases, around 20% were not identified and therefore not included. Additionally, the Cox regression did not adjust for other healthcare contacts that may have occured in the 72 hours following the index call, which may have affected the likelihood of a caller attending ED. We did intend on including primary care contact after the NHS 111 index call, but prior to ED attendance, as a covariate, but this resulted in a model which irredeemably violated the proportional hazards assumption and so was removed.

Despite mitigations aimed at resolving the proportional hazards assumption violation, we were unable to entirely avoid this, which may affect the accuracy of the results. However, reassuring log-log and Schoenfeld plots (Appendix 2) indicate this violation to be fairly minor, and suggest this may simply be a result of our large sample size and relatively high event rate.

While the Connected Bradford research database has great utility for researchers wishing to explore how patients traverse the wider healthcare system, it is restricted to a discrete geographical region in West Yorkshire, which may affect the generalisability of the results we have reported. Bradford is mainly an urban area and the 13th most deprived local authority in England (out of 333) based on IMD [14]. Future work including a larger population is warranted.

Finally, the study design and use of observational data means that the results presented should be interpreted as association and not casual inference.

## Conclusion

The time-to-event analysis found that callers to NHS 111 who do not wait to be triaged, are slower to attend ED with a non-avoidable cause than those who are triaged and are more likely to attend ED with an avoidable cause than triaged callers. This suggests that, for patients with a serious health problem that would be considered non-avoidable at ED, triaging by NHS 111 was associated with a reduced time to ED attendance. Future research is required to explore why some callers abandon calls, whether their conditions truly require emergency care, and how call-handling processes can be optimised to ensure equitable and timely access to appropriate services.

## Supporting information

**S1 Table. Modification to the original O'Keeffe et al non-avoidable Emergency Department admission criteria.**
(PDF)

**S1 File. Summary of final model for the EXPECT study.**
(PDF)

## Acknowledgments

The authors are grateful for the support and advice provided by Colin O'Keeffe and Dr. Susan Croft in relation to the avoidable ED attendance criteria and an early draft of this paper, and to Mike Bradburn for his support with the cox regression analysis.

## Author contributions

**Conceptualization:** Richard Pilbery.

**Data curation:** Richard Pilbery.

**Formal analysis:** Richard Pilbery.

**Methodology:** Richard Pilbery, Jen Lewis, Rebecca Simpson.

**Supervision:** Jen Lewis, Rebecca Simpson.

**Writing – original draft:** Richard Pilbery, Jen Lewis, Rebecca Simpson.

**Writing – review & editing:** Richard Pilbery, Jen Lewis, Rebecca Simpson.

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
