## [Decision Letter · Decision Letter 0]

13 Feb 2025

Dear Dr. Pilbery,

Thank you for submitting your manuscript to PLOS ONE. After careful consideration, we feel that it has merit but does not fully meet PLOS ONE’s publication criteria as it currently stands. Therefore, we invite you to submit a revised version of the manuscript that addresses the points raised during the review process.

Based on the Reviewer comments following are the key point you must consider during revison process. The detailed comments from the reviewers are available at the end.

The **title** should be revised to ensure it is more scientific and accurately reflects the study's focus and outcomes.should be revised to ensure it is more scientific and accurately reflects the study's focus and outcomes.The **methodology** section needs further details on the study design, including the specific variables assessed and data collection methods. Additionally, the statistical methods and software used should be explicitly mentioned.section needs further details on the study design, including the specific variables assessed and data collection methods. Additionally, the statistical methods and software used should be explicitly mentioned.The **statistical analysis** requires improvement. The authors should apply the requires improvement. The authors should apply the **Chi-square (χ²) test** to analyze the association between patient characteristics and call outcomes. to analyze the association between patient characteristics and call outcomes. **Table 1** should include call duration as either means with standard deviations or categorized values for better interpretation.should include call duration as either means with standard deviations or categorized values for better interpretation.The manuscript should ensure **clarity in terminology and abbreviations** by defining by defining **IMD** in the key notes of in the key notes of **Table 1** to enhance reader understanding.to enhance reader understanding.The **results** section should provide a more detailed explanation of the findings, ensuring they are well-supported by data and statistical analysis.section should provide a more detailed explanation of the findings, ensuring they are well-supported by data and statistical analysis.The **discussion and recommendations** should include practical suggestions on improving patient outcomes based on the study’s findings.should include practical suggestions on improving patient outcomes based on the study’s findings.The **ethics and acknowledgments** sections should explicitly state ethical considerations and disclose financial support sources, if applicable, to maintain transparency.sections should explicitly state ethical considerations and disclose financial support sources, if applicable, to maintain transparency.

We look forward to receiving your revised manuscript.

Kind regards,

Asim Mehmood

Academic Editor

PLOS ONE

2. 'Please provide additional details regarding participant consent. In the ethics statement in the Methods and online submission information, please ensure that you have specified (1) whether consent was informed and (2) what type you obtained (for instance, written or verbal, and if verbal, how it was documented and witnessed). If your study included minors, state whether you obtained consent from parents or guardians. If the need for consent was waived by the ethics committee, please include this information.

3. Thank you for stating the following financial disclosure:  [This report is independent research funded by the National Institute for Health and Care Research, Yorkshire and Humber Applied Research Collaborations NIHR200166. This study is also based on data from Connected Bradford (REC 18/YH/0200 & 22/EM/0127). The data is provided by the citizens of Bradford and district, and collected by the NHS, DfE and other organisations as part of their care and support. The views expressed in this publication are those of the author(s) and not necessarily those of the NHS, the National Institute for Health and Care Research or the Department of Health and Social Care.].  Please state what role the funders took in the study.  If the funders had no role, please state: "The funders had no role in study design, data collection and analysis, decision to publish, or preparation of the manuscript." If this statement is not correct you must amend it as needed. Please include this amended Role of Funder statement in your cover letter; we will change the online submission form on your behalf.

4. Please include a caption for figure 1.

Additional Editor Comments (if provided):

Reviewers' comments:

Reviewer's Responses to Questions

**Comments to the Author**

1. Is the manuscript technically sound, and do the data support the conclusions?

Reviewer #1: Yes

Reviewer #2: Yes

2. Has the statistical analysis been performed appropriately and rigorously?

Reviewer #1: No

Reviewer #2: No

3. Have the authors made all data underlying the findings in their manuscript fully available?

Reviewer #1: Yes

Reviewer #2: Yes

4. Is the manuscript presented in an intelligible fashion and written in standard English?

Reviewer #1: Yes

Reviewer #2: Yes

Reviewer #1: Please rewrite the title in more scientific way, for example " Outcomes of patients accessing NHS 111 calls"

In the methods please add details about the study design used and mention the variables that were assessed and how data were collected.

In the results, please use Chi 2 test to assess the association between the characteristics of the patients and the call outcome.

In table one please mention the length of the call in each group as means or categorically.

Please in the key note of table 1 explain what the abbreviation IMD is.

In the recommendation suggest how to improve the calls patients outcome.

Reviewer #2: In addition to thanking the authors, the following topics could be helpful to explain:

Abstract:

The method should better address the type of tests and software.

The findings could be stated in more detail

Ethics

Ethical considerations are mentioned in my paper and are not a specific issue

These should be mentioned

Acknowledgements

Financial sources

**Do you want your identity to be public for this peer review?** For information about this choice, including consent withdrawal, please see our For information about this choice, including consent withdrawal, please see our Privacy Policy .

Reviewer #1: **Yes:** Ghada Omer Hamad Abd El-RaheemGhada Omer Hamad Abd El-Raheem

Reviewer #2: No

While revising your submission, please upload your figure files to the Preflight Analysis and Conversion Engine (PACE) digital diagnostic tool, https://pacev2.apexcovantage.com/ . PACE helps ensure that figures meet PLOS requirements. To use PACE, you must first register as a user. Registration is free. Then, login and navigate to the UPLOAD tab, where you will find detailed instructions on how to use the tool. If you encounter any issues or have any questions when using PACE, please email PLOS at . PACE helps ensure that figures meet PLOS requirements. To use PACE, you must first register as a user. Registration is free. Then, login and navigate to the UPLOAD tab, where you will find detailed instructions on how to use the tool. If you encounter any issues or have any questions when using PACE, please email PLOS at figures@plos.org . Please note that Supporting Information files do not need this step.. Please note that Supporting Information files do not need this step.

---

## [Author Response · Author response to Decision Letter 1]

17 Apr 2025

Reviewer comment: The title should be revised to ensure it is more scientific and accurately reflects the study's focus and outcomes.

Author’s response: Title updated.

Reviewer comment: The methodology section needs further details on the study design, including the specific variables assessed and data collection methods. Additionally, the statistical methods and software used should be explicitly mentioned.

Author’s response: Addition of the use of R and specific time-to-event analysis tools added. Specific variables are stated in the primary analysis section of the methods.

Reviewer comment: The statistical analysis requires improvement. The authors should apply the Chi-square (χ²) test to analyze the association between patient characteristics and call outcomes.

Author’s response: It is the view of the statisticians that Chi-square tests i.e. univariate testing are not necessary or informative since the table is describing the cohort and important covariates are already controlled in the Cox analysis.

Reviewer comment: Table 1 should include call duration as either means with standard deviations or categorized values for better interpretation.

Author’s response: Call duration was not available in the data and so has not been presented. The other continuous variables have been presented as median and inter-quartile ranges given that the data was not normally distributed and mean/standard deviation was not appropriate.

Reviewer comment: The manuscript should ensure clarity in terminology and abbreviations by defining IMD in the key notes of Table 1 to enhance reader understanding.

Author’s response: The abbreviations used in the table were provided in full in the text prior to table 1. However, IMD and ED have been written in full in table 1 to improve clarity as requested.

Reviewer comment: The results section should provide a more detailed explanation of the findings, ensuring they are well-supported by data and statistical analysis.

Author’s response: We are struggling to know what is being requested here without repeating the contents of the tables. We would appreciate some specific advice as to what the reviewers feel is missing.

Reviewer comment: The discussion and recommendations should include practical suggestions on improving patient outcomes based on the study’s findings.

Author’s response: Additional paragraph added to the Discussion section to address this.

Reviewer comment: The ethics and acknowledgments sections should explicitly state ethical considerations and disclose financial support sources, if applicable, to maintain transparency.

Author’s response: Ethics section updated.

Reviewer comment: Please ensure that your manuscript meets PLOS ONE's style requirements, including those for file naming. The PLOS ONE style templates can be found at

Author’s response: Requirements reviewed and manuscript updated to conform.

Reviewer comment: Please provide additional details regarding participant consent. In the ethics statement in the Methods and online submission information, please ensure that you have specified (1) whether consent was informed and (2) what type you obtained (for instance, written or verbal, and if verbal, how it was documented and witnessed). If your study included minors, state whether you obtained consent from parents or guardians. If the need for consent was waived by the ethics committee, please include this information.

Author’s response: Ethics section updated to highlight that the research database already has the necessary ethical approvals for the use of patient data for research purposes without the need for individual patient consent.

Note that this study only involved adults.

Reviewer comment: If you are reporting a retrospective study of medical records or archived samples, please ensure that you have discussed whether all data were fully anonymized before you accessed them and/or whether the IRB or ethics committee waived the requirement for informed consent. If patients provided informed written consent to have data from their medical records used in research, please include this information.

Author’s response: The Data section explains this. Ethics section has been updated.

Reviewer comment: Thank you for stating the following financial disclosure: [This report is independent research funded by the National Institute for Health and Care Research, Yorkshire and Humber Applied Research Collaborations NIHR200166. This study is also based on data from Connected Bradford (REC 18/YH/0200 & 22/EM/0127). The data is provided by the citizens of Bradford and district, and collected by the NHS, DfE and other organisations as part of their care and support. The views expressed in this publication are those of the author(s) and not necessarily those of the NHS, the National Institute for Health and Care Research or the Department of Health and Social Care.]. Please state what role the funders took in the study. If the funders had no role, please state: "The funders had no role in study design, data collection and analysis, decision to publish, or preparation of the manuscript." If this statement is not correct you must amend it as needed. Please include this amended Role of Funder statement in your cover letter; we will change the online submission form on your behalf.

Author’s response: Statement updated as requested.

Reviewer comment: We note that you have indicated that there are restrictions to data sharing for this study. For studies involving human research participant data or other sensitive data, we encourage authors to share de-identified or anonymized data. However, when data cannot be publicly shared for ethical reasons, we allow authors to make their data sets available upon request. Before we proceed with your manuscript, please address the following prompts:

b) If there are no restrictions, please upload the minimal anonymized data set necessary to replicate your study findings to a stable, public repository and provide us with the relevant URLs, DOIs, or accession numbers.

Author’s response: Data sharing statement has been updated on the portal.

Reviewer comment: Please include a caption for figure 1.

Author’s response: The caption for Figure 1 is on page 11 of the manuscript.

---

## [Decision Letter · Decision Letter 1]

16 Jul 2025

Dear Dr. Pilbery,

plosone@plos.org . . A rebuttal letter that responds to each point raised by the academic editor and reviewer(s). You should upload this letter as a separate file labeled 'Response to Reviewers'.A marked-up copy of your manuscript that highlights changes made to the original version. You should upload this as a separate file labeled 'Revised Manuscript with Track Changes'.An unmarked version of your revised paper without tracked changes. You should upload this as a separate file labeled 'Manuscript'.

We look forward to receiving your revised manuscript.

Kind regards,

Tai-Heng Chen, M.D., Ph.D.

Academic Editor

PLOS ONE

Journal Requirements:

Reviewers' comments:

Reviewer's Responses to Questions

**Comments to the Author**

Reviewer #2: (No Response)

2. Is the manuscript technically sound, and do the data support the conclusions?

Reviewer #2: Partly

3. Has the statistical analysis been performed appropriately and rigorously?

Reviewer #2: No

4. Have the authors made all data underlying the findings in their manuscript fully available?

Reviewer #2: Yes

5. Is the manuscript presented in an intelligible fashion and written in standard English?

Reviewer #2: Yes

Reviewer #2: Dear Authors,

Due to the review, changes in the text of the article have not been highlighted.

If there are any changes, please mention them.

During the review, the acknowledgements and thanks have been specified at the end.

The statement of the ethics committee has been included.

The title has been revised.

References should be included in the journal according to the journal's guidelines.

The discussion of this article can better review the studies

**Do you want your identity to be public for this peer review?** For information about this choice, including consent withdrawal, please see our For information about this choice, including consent withdrawal, please see our Privacy Policy .

Reviewer #2: No

While revising your submission, please upload your figure files to the Preflight Analysis and Conversion Engine (PACE) digital diagnostic tool, https://pacev2.apexcovantage.com/ . PACE helps ensure that figures meet PLOS requirements. To use PACE, you must first register as a user. Registration is free. Then, login and navigate to the UPLOAD tab, where you will find detailed instructions on how to use the tool. If you encounter any issues or have any questions when using PACE, please email PLOS at . PACE helps ensure that figures meet PLOS requirements. To use PACE, you must first register as a user. Registration is free. Then, login and navigate to the UPLOAD tab, where you will find detailed instructions on how to use the tool. If you encounter any issues or have any questions when using PACE, please email PLOS at figures@plos.org . Please note that Supporting Information files do not need this step.. Please note that Supporting Information files do not need this step.

---

## [Author Response · Author response to Decision Letter 2]

27 Aug 2025

2. Is the manuscript technically sound, and do the data support the conclusions?

Reviewer #2: Partly

We have fully addressed the reviewer’s comments in the previous revision. If the reviewer can provide comprehensive and verbose feedback, then we would be happy to address any outstanding issues.

3. Has the statistical analysis been performed appropriately and rigorously?

Reviewer #2: No

Two of the authors are professional statisticians and are satisfied that within the limitations of the observational data utilised for the study, the statistical analysis has been performed appropriate and rigorously.

Due to the review, changes in the text of the article have not been highlighted.

I include this for the attention of the editorial team since it would be helpful for reviewer’s to see the tracked change submission we provided.

During the review, the acknowledgements and thanks have been specified at the end.

Noted.

The statement of the ethics committee has been included.

Noted.

The title has been revised.

Noted.

References should be included in the journal according to the journal's guidelines.

References updated as requested.

The discussion of this article can better review the studies.

We have deliberately not done this since the studies are only tangentially related to the current study, and we did not want to bring any further attention to them. As we state in the manuscript, as far as we are aware, there is no research specifically looking at addressing our research question.

---

## [Decision Letter · Decision Letter 2]

4 Nov 2025

Dear Dr. Pilbery,

We look forward to receiving your revised manuscript.

Kind regards,

Mergan Naidoo, PhD

Academic Editor

PLOS ONE

Journal Requirements:

Reviewers' comments:

Reviewer's Responses to Questions

**Comments to the Author**

Reviewer #2: (No Response)

Reviewer #3: (No Response)

2. Is the manuscript technically sound, and do the data support the conclusions?

Reviewer #2: Partly

Reviewer #3: Partly

3. Has the statistical analysis been performed appropriately and rigorously?

Reviewer #2: Yes

Reviewer #3: Yes

4. Have the authors made all data underlying the findings in their manuscript fully available?

Reviewer #2: Yes

Reviewer #3: Yes

5. Is the manuscript presented in an intelligible fashion and written in standard English?

Reviewer #2: Yes

Reviewer #3: Yes

Reviewer #2: Dear authors,

The introduction could be better explained. During the review, the necessity of this study and the innovative aspects could play a role in highlighting the objectives of this study.

In the discussion of the study:

In the case of the first study Patient compliance with NHS 111 advice: Analysis of adult call and ED attendance data 2013–2017, could it be similar to your study? If it is similar, the introduction of the discussion should be reviewed. Considering the authors' response that you mentioned a limited study in the discussion. Using similar studies can address the necessity of this research and the study gap, which plays an important role in explaining your study.

If there is a change in the text, it should be clearly indicated.

Reviewer #3: Thank you for the opportunity to review this important and well-written manuscript. It explores a relevant and under-studied aspect of urgent care utilisation, focusing on the outcomes of callers who end NHS 111 calls before triage.

The manuscript is clearly organised, and the use of the Connected Bradford dataset adds valuable depth to the analysis.

I have suggestions aimed at improving clarity, flow, and methodological transparency. These comments are offered in a constructive spirit to support the authors in strengthening the paper’s interpretation and ensuring that the conclusions remain well aligned with the evidence presented.

Abstract:

Replace any causal language with associational phrasing. For example, instead of stating that triaging “supports patients to seek appropriate help more quickly”, state that triaging “was associated with shorter times to emergency department attendance”.

Correct the typographical error “whot” to “who”.

End the abstract with a balanced statement acknowledging that the study design does not allow causal inference.

Introduction:The introduction is useful and well written. It provides clear context on NHS 111 and explains why understanding call abandonment is an important area for research. The topic is timely and relevant to health service policy and patient safety.

The methods section is generally well described and demonstrates thoughtful use of the Connected Bradford dataset. The inclusion of linked data sources is a major strength. The overall structure is logical, and the statistical approach is appropriate for the research question.

Stats: Justify the choice of a seventy-two-hour window for emergency department attendance and a thirty-second threshold for defining an abandoned call. These thresholds appear arbitrary and may influence the observed association

Discussion:

The discussion is thoughtful and covers several relevant points, but at present it over-interprets the findings and gives too little attention to alternative explanations. The tone suggests causation where only association can be inferred. I believes strengthening this section will make the interpretation more balanced and credible.

a. Reframe all causal statements as associations. For instance, replace “triaging helps patients to seek appropriate care sooner” with “triaging was associated with earlier emergency department attendance”. This will ensure the interpretation remains aligned with the observational nature of the study.

b.Remove or substantially revise the comparison with patients who leave the emergency department without being seen. These represent a different decision-making context and do not provide a valid analogy for telephone triage abandonment.

Limitations: Mention that the study period included times of exceptional NHS strain, including winter pressures and post-pandemic recovery, which may have affected both call volumes and caller behaviour.

Conclusion: The conclusion is concise and logically follows the results, but the current wording implies causation and certainty that the study design cannot support. Reiterate that the data cannot determine whether call abandonment represents a risk to patients or a reflection of less urgent needs. This distinction is critical and should be acknowledged directly. The study is observational and cannot establish causation. Phrases such as:

“triaging by NHS 111 supports those patients to seek appropriate help more quickly”

“NHS 111 may help patients who do not require urgent care from attending an ED unnecessarily”

imply cause and effect — suggesting that triage made people attend sooner or prevented unnecessary attendance.

In reality, the analysis can only show associations. Callers who stayed on the line for triage might differ systematically from those who abandoned the call (for example, in urgency, health literacy, or motivation). So the findings show a relationship, not an effect.

Line 264: “whot” should be corrected to “who”.

I also believe that a brief statement about future research should be included within the conclusion. The current ending summarises the findings well but would benefit from a forward-looking sentence that highlights the next steps. I suggest adding a line such as:

“Future research should explore why some callers abandon calls, whether their conditions truly require emergency care, and how call-handling processes can be optimised to ensure equitable and timely access to appropriate services.”

This is a valuable and timely piece of research that adds to our understanding of urgent care utilisation. The suggestions provided are intended to strengthen the paper’s clarity and ensure that its interpretations remain appropriately balanced and transparent. I hope the authors will receive them in the spirit of refining an already high-quality piece of work.

**Do you want your identity to be public for this peer review?** For information about this choice, including consent withdrawal, please see our For information about this choice, including consent withdrawal, please see our Privacy Policy .

Reviewer #2: No

Reviewer #3: No

You may also use PLOS’s free figure tool, NAAS, to help you prepare publication quality figures: https://journals.plos.org/plosone/s/figures#loc-tools-for-figure-preparation

---

## [Author Response · Author response to Decision Letter 3]

25 Mar 2026

Reviewer comment:

The introduction could be better explained. During the review, the necessity of this study and the innovative aspects could play a role in highlighting the objectives of this study.

Author’s response:

We feel that the current background section is appropriate and does not delay the reader from getting ‘into’ the paper.

Reviewer comment:

The introduction is useful and well written. It provides clear context on NHS 111 and explains why understanding call abandonment is an important area for research. The topic is timely and relevant to health service policy and patient safety.

Author’s response:

Thank you.

Reviewer comment:

In the case of the first study Patient compliance with NHS 111 advice: Analysis of adult call and ED attendance data 2013–2017, could it be similar to your study?

Author’s response:

No, this study only examined successfully triaged calls and did not contain any follow-up involving primary care services.

Reviewer comment:

Considering the authors' response that you mentioned a limited study in the discussion. Using similar studies can address the necessity of this research and the study gap, which plays an important role in explaining your study.

Author’s response:

There are no other comparable studies, which made the discussion more challenging with respect to its context in the wider literature.

Reviewer comment:

Remove or substantially revise the comparison with patients who leave the emergency department without being seen. These represent a different decision-making context and do not provide a valid analogy for telephone triage abandonment.

Author’s response:

Removed the comparison as suggested.

Reviewer comment:

Replace any causal language with associational phrasing. For example, instead of stating that triaging “supports patients to seek appropriate help more quickly”, state that triaging “was associated with shorter times to emergency department attendance”.

Author’s response:

Thank you for the suggested wording change. Manuscript updated.

Reviewer comment:

Correct the typographical error “whot” to “who”.

Author’s response:

Spelling corrected.

Reviewer comment:

End the abstract with a balanced statement acknowledging that the study design does not allow causal inference.

Author’s response:

We have changed the language of the final statement that should make it clear to the reader that the results are not based on causal inference. However, we had added a specific sentence in the limitations section to reflect this suggestion.

Reviewer comment:

The methods section is generally well described and demonstrates thoughtful use of the Connected Bradford dataset. The inclusion of linked data sources is a major strength. The overall structure is logical, and the statistical approach is appropriate for the research question.

Author’s response:

Thank you.

Reviewer comment:

Stats: Justify the choice of a seventy-two-hour window for emergency department attendance and a thirty-second threshold for defining an abandoned call. These thresholds appear arbitrary and may influence the observed association.

Author’s response:

72 hours was deemed by local expert opinion to be a reasonable time period to have some confidence that the reason for the 111 call and ED attendance were related. The greater the time period, the more tenuous the relationship was thought to be.

The 30 second threshold was chosen as NHS England differentiates between calls that end at 30 seconds or less in reporting requirements for NHS 111 service performance. In addition, calls lasting less than 30 seconds are more likely to be made in error in our view.

Reviewer comment:

Reframe all causal statements as associations. For instance, replace “triaging helps patients to seek appropriate care sooner” with “triaging was associated with earlier emergency department attendance”. This will ensure the interpretation remains aligned with the observational nature of the study.

Author’s response:

Manuscript updated.

Reviewer comment:

Limitations: Mention that the study period included times of exceptional NHS strain, including winter pressures and post-pandemic recovery, which may have affected both call volumes and caller behaviour.

Author’s response:

We disagree. The time period of this study (and indeed up to the present day) suggests that the current volume and state of the NHS is the ‘new’ normal. As such, we think our results are representative of the post-pandemic state of the NHS, which is likely to remain for some time to come.

Reviewer comment:

I also believe that a brief statement about future research should be included within the conclusion. The current ending summarises the findings well but would benefit from a forward-looking sentence that highlights the next steps. I suggest adding a line such as:

“Future research should explore why some callers abandon calls, whether their conditions truly require emergency care, and how call-handling processes can be optimised to ensure equitable and timely access to appropriate services.”

Author’s response:

Thank you, manuscript updated with the suggestion.

Reviewer comment:

This is a valuable and timely piece of research that adds to our understanding of urgent care utilisation. The suggestions provided are intended to strengthen the paper’s clarity and ensure that its interpretations remain appropriately balanced and transparent. I hope the authors will receive them in the spirit of refining an already high-quality piece of work.

Author’s response:

Thank you.

---

## [Editor Report · Decision Letter 3]

26 Mar 2026

Emergency department attendance by callers to NHS111 who end the call prior to triage: A time-to-event-analysis

PONE-D-24-55555R3

Dear Dr. Richard Pilbery

We’re pleased to inform you that your manuscript has been judged scientifically suitable for publication and will be formally accepted for publication once it meets all outstanding technical requirements.

Kind regards,

Mergan Naidoo, PhD

Academic Editor

PLOS One
---

## [Editor Report · Acceptance letter]

PONE-D-24-55555R3

PLOS One

Dear Dr. Pilbery,

I'm pleased to inform you that your manuscript has been deemed suitable for publication in PLOS One. Congratulations! Your manuscript is now being handed over to our production team.

Kind regards,

on behalf of

Professor Mergan Naidoo

Academic Editor

PLOS One